# PhysioKit: An Open-Source, Low-Cost Physiological Computing Toolkit for Single- and Multi-User Studies

**DOI:** 10.3390/s23198244

**Published:** 2023-10-04

**Authors:** Jitesh Joshi, Katherine Wang, Youngjun Cho

**Affiliations:** Department of Computer Science, University College London, London NW1 2AE, UK; jitesh.joshi.20@ucl.ac.uk (J.J.); katherine.wang.19@ucl.ac.uk (K.W.)

**Keywords:** physiological computing, data acquisition toolkit, multi-user HCI studies, biofeedback, signal quality assessment

## Abstract

The proliferation of physiological sensors opens new opportunities to explore interactions, conduct experiments and evaluate the user experience with continuous monitoring of bodily functions. Commercial devices, however, can be costly or limit access to raw waveform data, while low-cost sensors are efforts-intensive to setup. To address these challenges, we introduce *PhysioKit*, an open-source, low-cost physiological computing toolkit. *PhysioKit* provides a one-stop pipeline consisting of (i) a sensing and data acquisition layer that can be configured in a modular manner per research needs, and (ii) a software application layer that enables data acquisition, real-time visualization and machine learning (ML)-enabled signal quality assessment. This also supports basic visual biofeedback configurations and synchronized acquisition for co-located or remote multi-user settings. In a validation study with 16 participants, *PhysioKit* shows strong agreement with research-grade sensors on measuring heart rate and heart rate variability metrics data. Furthermore, we report usability survey results from 10 small-project teams (44 individual members in total) who used *PhysioKit* for 4–6 weeks, providing insights into its use cases and research benefits. Lastly, we discuss the extensibility and potential impact of the toolkit on the research community.

## 1. Introduction

Physiological signals have been actively explored in a wide range of research fields, given their usefulness in tracking health, as well as physical and psychological states. Particularly in human–computer interaction (HCI), much attention has recently been paid to physiological computing research, where it focuses on how to evaluate user responses to interventions, enhance user interactions with technology, or investigate how to support interpersonal interactions [1]. Physiological computing involves detecting, acquiring, and processing various physiological signals, such as cardiac rhythm or heart rate (HR), skin conductance, blood-oxygen saturation, body temperature, blood glucose levels, muscle activity, and neural activity. Sensors commonly used for acquisition of physiological signals include: electrocardiogram (ECG), photoplethysmography (PPG), respiratory (RSP), electrodermal activity (EDA), electromyography (EMG) and electroencephalogram (EEG) [2]. The functions of physiological computing systems include acquisition, interpretation, and facilitation of different schemes of interactions, as well as personalized interventions [3,4,5].

While the proliferation of low-cost consumer-grade physiological sensing solutions and fitness trackers (e.g., Apple Watch, https://www.apple.com/uk/watch/, accessed on 28 September 2023, Fitbit, https://www.fitbit.com/, accessed on 28 September 2023, Garmin, https://www.garmin.com/, accessed on 28 September 2023) has boosted the interest in physiological sensing forward, there is a need to pay attention to the challenges of using such devices, including limitations in robustness associated with sensor misplacement, body movements and ambient noise, reduced flexibility in adapting an acquisition interface to the study protocol, and relatively high costs of hardware or software [6,7,8,9]. The majority of these consumer-grade sensors provide limited access to continuous physiological signals and metrics (e.g., inter-beat interval time-series), which can yield considerable insight into our bodily functions and psychological states [8,10,11,12]. Data acquired from these devices is processed using the manufacturer’s proprietary algorithms and exported directly to remote company servers for commercial purposes [13,14], raising ethical concerns about data privacy. Furthermore, the rapid release cycles of new commercial wearable sensor models pose additional research challenges. While the general public may assume that newer models can perform better, it is difficult to conduct and publish validation studies on the same timescale of of new technology releases.

Research-grade physiological sensing devices (e.g., Procomp Infiniti System (https://thoughttechnology.com/procomp-infiniti-system-w-biograph-infiniti-software-t7500m/ (accessed on 28 September 2023)) from Thought Technology, BIOPAC (https://www.biopac.com/(accessed on 28 September 2023)), Empatica (https://www.empatica.com/(accessed on 28 September 2023)), Biofourmis (https://biofourmis.com/(accessed on 28 September 2023)) are often expensive, leading to wide adoption of affordable physiological sensing solutions (e.g., BITalino (https://www.pluxbiosignals.com/collections/bitalino(accessed on 28 September 2023)), OpenBCI (https://openbci.com/(accessed on 28 September 2023)) by the research community. Owing to the access to raw signal data, the research community has demonstrated several application use-cases of research-grade devices and toolkits, including OpenBCI, https://openbci.com/citations (accessed on 28 September 2023), Empatica E4, https://www.empatica.com/research/publications/ (accessed on 28 September 2023), BITalino, https://scholar.google.com/scholar?as_sdt=0,5&q=BITalino&hl=en&as_ylo=2000&as_yhi=2021 (accessed on 28 September 2023) and Movisens GmbH, https://www.movisens.com/en/resources/publications/ (accessed on 28 September 2023), and has contributed towards their validation. Researchers have also explored open-source prototyping platforms (e.g., Arduino with physiological sensing nodes (https://www.arduino.cc/ (accessed on 28 September 2023))) in different contexts [15,16,17,18]. Physiological computing with such platforms often requires a range of technical and computational skills and considerable setup time. Additionally, though few open-source platforms and affordable toolkits are sufficiently validated across the entire healthy-range of physiological variations [19], the usability study of the interface for physiological signal acquisition is often neglected.

Signal quality is another concerning factor both for consumer-grade as well as research-grade devices, which is affected by environmental and experimental factors. Motion artifact noise, as well as participant non-compliance from discomfort while wearing such sensors in different scenarios or physical impairments, may result in a reduced quality of acquired signals [20,21]. While these challenges cannot be fully addressed, one of the potential ways to circumvent these issues can be to enable flexibility in sensor placement (e.g., PPG sensors can be placed on either ear-lobes or fingers). This flexibility also enhances the accessibility of the sensors to suit users with different physical sensitivities or requirements. The complementary measure can be a provision for real-time assessment of signal quality to flag noisy acquisition fragments of signals [22], which would enable researchers to take appropriate actions during data acquisition or processing. Physiological sensing toolkits often neglect such a feature. Lastly, due to social distancing needs in recent years, studies have been conducted in remote settings [23]. While several existing toolkits can support synchronized acquisition from multiple co-located users, provision for synchronized acquisition of physiological signals for remote multi-user scenarios is yet not addressed.

It is therefore crucial to design a toolkit that considers the above mentioned design objectives and existing research gaps. These include access to raw data, provision for synchronized data acquisition in remotely located multi-user scenarios, support for real-time analysis of signal for physiological metrics, as well as signal quality assessment, provision for configuring experimental protocols, and a provision to transmit real-time analysis data for adaptive interactions as well as customized interventions. To our best knowledge, there is no open-source platform for physiological computing, which encompasses all the design objectives on a single platform. To address this, we propose *PhysioKit*, a new open-source physiological computing toolkit for HCI researchers, hobbyists and practitioners. While being cost-effective, *PhysioKit* can facilitate flexible experiment configuration, data collection, and support real-time analyses of physiological data. Our contribution is two-fold:A novel open-source physiological computing toolkit (GitHub repo link, https://github.com/PhysiologicAILab/PhysioKit, accessed on 28 September 2023) that offers a one-stop physiological computing pipeline spanning from data collection and processing to a wide range of analysis functions, including a new machine learning module for the physiological signal quality assessment;A report on validation study results, as well as user reports on the *PhysioKit*’s usability and examples of use cases demonstrating its applicability in diverse applications.

We begin this paper with an overview of related work and identify key challenges. We then introduce *PhysioKit*, describing its sensing and software application layers, and how they are designed to enable high-quality data acquisition, while considering the flexibility in selecting physiological sensors. We present results from a validation study, an overview of applications in which *PhysioKit* is used and outcomes from a usability survey. Finally, we conclude with discussions on the implications and benefits of *PhysioKit* for different contexts and user communities.

## 2. Related Work

In this section, we first discuss prior work on applications of physiological computing in HCI research and then review existing physiological computing devices and toolkits related to the contribution we make in this work.

### 2.1. Physiological Computing and HCI Applications

The applications of physiological computing in HCI contexts can be broadly categorized into two themes, based on the role physiological computing plays in the interaction: *interventional* and *passive*. This categorization serves as one of the key design considerations for the development of physiological computing toolkits. For interventional studies, real-time computing of physiological metrics is required, which can further be mapped to biofeedback design or to adapt the interaction with the self, with others, or with technology. On the other hand, for passive physiological computing studies, the provision of data acquisition meets the research need, as these studies do not require adapting any interaction aspects based on real-time computing of physiological metrics from the acquired signals.

Interventional studies have examined how real-time physiological computing can be used in several contexts, including health monitoring (e.g., stress [24], diabetes [25]), training healthful practices (e.g., respiration [26]), educating children in anatomy [27], communicating affective states between people during chats [28] or VR gameplay [29], sensing passenger comfort in smart cars [30,31], and personalizing content through adaptive narratives (e.g., interactive storytelling [32], adjustable cultural heritage experiences [33], synchronized content between multiple users [34]). Illustrative studies with passive use of physiological computing include assessing user’s mental states (e.g., stress, workload and attention) [5,10,35,36], exploring user experiences [37,38,39,40], or objective comparisons with subjective reports [41]. The illustrated HCI studies are not exhaustive and are mentioned to emphasize the two distinct ways in which researchers use physiological computing. While most commercial research-grade and low-cost devices support data acquisition for passive studies, they are often not designed considering different needs of interventional studies.

### 2.2. Physiological Computing Sensors, Devices and Toolkits

PPG, EDA and RSP are among the most prominent physiological sensing channels used in HCI research [1,42,43]. PPG and EDA signals, for instance, have been explored to capture physiological and emotional states [44], engagement levels [45,46], for communicating emotional states [47,48], as well for evoking empathy through shared biofeedback [49]. Respiration cues have been used to help people understand and manage their stress [50,51], or to feel connected to others through sharing breathing signals [52]. To acquire these signals, researchers generally consider consumer-grade devices acceptable to use and prioritize the use of devices based on design and familiarity with the brand, rather than reliability, as well as comfort or ease of use [53]. Here, we present a brief overview of the data acquisition devices as well as data analysis toolkits.

#### 2.2.1. Sensing and Data Acquisition Devices

##### Support for Passive and Interventional HCI Studies

For passive HCI studies (Section 2.1), researchers can choose from broader spectrum of devices ranging from expensive research-grade physiological sensing devices, including Procomp Infiniti, https://thoughttechnology.com/procomp-infiniti-system-w-biograph-infiniti-software-t7500m/ (accessed on 28 September 2023), BIOPAC, https://www.biopac.com/ (accessed on 28 September 2023), Shimmer, https://shimmersensing.com/product/consensys-bundle-development-kit/ (accessed on 28 September 2023), Empatica, https://www.empatica.com/ (accessed on 28 September 2023), and Biofourmis, https://biofourmis.com/ (accessed on 28 September 2023), to more affordable devices including BITalino, https://www.pluxbiosignals.com/collections/bitalino (accessed on 28 September 2023), OpenBCI, https://openbci.com/ (accessed on 28 September 2023) and Movisens GmbH, https://www.movisens.com/en/resources/publications/ (accessed on 28 September 2023), among several others. However, it can be observed that for interventional studies (Section 2.1), researchers often combine open-source or affordable sensing hardware with their custom developed software [52,54]. While affordable sensing toolkits offer greater flexibility for using their sensing hardware, they insufficiently support configuration for experiments, as well as several types of biofeedback modalities with real-time signal analysis. This requires researchers to spend significant efforts towards customizing the acquisition pipeline with real-time analysis and bio-feedback presentation.

##### Real-Time Signal Quality Assessment

The signal quality of contact-based physiological sensors get affected in presence of relative motion between sensor and skin surface. PPG sensors, for instance, are susceptible to motion that results in interference from natural and artificial light [55], as well as to varying pressure at the sensor site caused by the activities of daily living affecting the blood flow [56]. Factors leading to artifacts cannot be controlled, though it is often possible to perform signal quality assessment and eliminate the noisy segments from the analysis. While signal quality assessment for physiological signals is an active research field [22,57,58,59,60,61], existing physiological sensing solutions do not offer provision of assessing signal quality, which can immensely increase validity of the analysis. Widely used methods for signal quality assessment include signal-to-noise ratio (SNR), template matching, and relative power signal quality index (pSQI) [61], along with recent machine learning-based approaches based on SVM classifier [60], LSTM [59], and 1D-CNN [57]. The state-of-the-art (SOTA) performance has been demonstrated by 1D-CNN classifier approach [57] achieving 0.978 accuracy on the MIMIC III PPG waveform database. It is noteworthy that stringent signal quality assessment may lower the signal retention, thereby decreasing the usable segments of signal for deriving physiological metrics [62] and, therefore, it is crucial to develop an optimal signal quality measure with an objective to minimize for false positives and false negatives.

##### Support for Remote Multi-User Studies

Furthermore, existing toolkits offer limited support for multi-user studies. For scenarios in which multi-user studies are conducted with remotely located users, researchers have either used time-stamping information [63] or have deployed manual approaches [64] for synchronizing the time-series data. These approaches of synchronizing the data acquisition are not suitable for interventional studies and may result in a varied amount of time-lag between the physiological signals of different users. One very recent work proposed an open-source toolkit [19] for synchronized acquisition of multiple physiological signals using their sensor fusion unit (SFU). However, it remains unclear if signal acquisition can be synchronized for multiple SFUs that are located at remote locations.

##### Validation Studies

Research-grade devices undergo rigorous performance tests, as they are required to adhere to the regulatory standards. In contrast, owing to their intended use, consumer-grade devices are not required to comply with medical regulatory standards. However, researchers have contributed to the validation of consumer-grade as well as open-source devices. We present a non-exhaustive scoping review with few of the these devices, along with the comparative analysis based on the published validation studies in Appendix A. For effectively validating the sensing devices, it is essential to induce sufficient variations in psychophysiological states of participants, such as the deployment of Stroop test [65] for validating open-source toolkit [19] as well the deployment of light-to-vigorous physical activity for validating wrist-worn wearables [66,67].

#### 2.2.2. Data Analysis Toolkits

Data analysis software are either bundled with the research-grade physiological sensing solutions, or these are available as add-on packages which are required to purchase separately for analyzing the physiological data [68,69,70,71]. Except for the open-source toolkits such as [19,70], commercial data analysis toolkits implement proprietary algorithms and offer limited flexibility in choosing the algorithms for computing metrics from physiological signals. In spite of advancing research towards analysis of physiological signals, the limited flexibility restricts the use of the state-of-the-art algorithms for computing physiological metrics. In past few years, open-source toolkits have emerged enabling researchers to process raw sensor data in more customized manner. A few notable examples among these toolkits are NeuroKit2 [72], HeartPy [73], FLIRT [74], and PyPhysio [75]. HeartPy [73] focuses primarily on PPG signals, while FLIRT [74], NeuroKit2 [72] and PyPhysio [75] support analyzing multiple physiological signals. Specifically, FLIRT [74] supports analyzing ECG, EDA and accelerometer signals, PyPhysio [75] adds support for PPG, EEG, EMG and RSP, while NeuroKit2 [72] further adds support for analyzing EOG, with each latter toolkit supporting the analysis of signals supported by former toolkits. With growing adoption of the Python programming language, all above-mentioned data analysis toolkits are available as Python libraries. Owing to the low computational complexity of the algorithms implemented by the toolkits, and the increasing availability of computational resources personal computers, these toolkits can be leveraged both for post-processing of acquired signals as well as for real-time computing of physiological metrics. Thus, an integrated solution that can be compatible with existing data analysis packages can immensely benefit interventional studies, providing simultaneous data acquisition and real-time analysis, which we address in this work.

## 3. The Proposed Physiological Computing Toolkit: PhysioKit

Figure 1 details the system architecture of *PhysioKit*. The toolkit consists of two layers: (i) the sensing and signal acquisition layer, and (ii) the software application layer. The former layer enables a modular setup for a wider range of physiological sensors which can be connected with a widely available micro-controller, thereby offering flexibility in configuring physiological sensors depending on the research needs. The software application layer includes a data collection module, a real-time streaming module, signal quality assessment module and a data analysis module. Our software application is built with Python, which can support different operating systems (e.g., Windows, Linux).

### 3.1. Sensing and Signal Acquisition Layer

This layer is designed to facilitate the flexibility of sensor placement, and provide options to configure the toolkit for various single and multi-user study settings. The sensing and signal acquisition layer of *PhysioKit* supports multiple inexpensive physiological sensors (e.g., PPG channels) that are compatible with a micro-controller (i.e., Arduino board as a default). These sensors can be connected to analog inputs of the board (e.g., A0–A3 pins in Arduino). Users can easily configure acquisition parameters (e.g., sampling rate and analog-to-digital conversion resolution). The layer transmits the collected sensor data to a connected computing device (e.g., laptop) via wired (USB) or wireless (Bluetooth) communication.

Table 1 lists some of the key parameters of the sensing and signal acquisition layer of *PhysioKit*, as well as it also mentions quick installation step and links to package contents. The Arduino board governs the hardware specifications, which can be selected according to research needs. For instance, some of the most cost-effective microcontroller boards (e.g., Arduino Uno, Arduino Nano) can support up to six physiological sensors and a sampling rate of up to 512 samples per second, whereas Arduino Due and Arduino Mega can support 12 and 16 channels, respectively.

To support flexible placement of physiological sensors, we designed an example template of a 3D-printable CAD model for a mountable PPG sensor wristband, as shown in Figure 2. This lets users acquire PPG signals from different body parts simultaneously, which can possibly help handle motion artifacts. The template model is available in the *PhysioKit* repository.

In this work, we focused on demonstrating the functionality and validation of the toolkit using two PPG sensors (https://pulsesensor.com/ (accessed on 28 September 2023)) connected to Arduino Due, which is presented in Section 4.1. There, we focus on validating HR and HRV measurements from PPG signal, as it is the most widely used physiological sensing channel. However, the interface and toolkit can be easily extended to other sensing channels, including RSP, EDA, ECG, EMG and EEG. To demonstrate the readiness of extending other sensors, we also integrated a RSP sensor (https://www.pluxbiosignals.com/products/respiration-pzt (accessed on 28 September 2023)) and an EDA sensor (https://seeeddoc.github.io/Grove-GSR_Sensor/(accessed on 28 September 2023)). Though the measurements of these alternative sensing channels were not validated in the current work, we discuss future plans to implement these in Figure 1.

### 3.2. Software Application Layer

The software application layer includes the user interface (UI), as depicted in Figure 3A, a real-time streaming and signal quality assessment module, as well as a data analysis module. The challenges concerning access to raw, unfiltered physiological signals and flexible configurations for passive and interventional studies involving both co-located and remote multi-user settings were carefully considered when developing the UI. The resulting interface includes features to facilitate data acquisition, signal visualization, and signal quality assessment, as well as bio-feedback visualization. The data analysis module builds upon existing data analysis toolkits [72,73] to further streamline analysis of physiological signals, as per the experimental protocol. The software module is implemented using the Python programming language, as it is a multi-platform language.

#### 3.2.1. User Interface

To optimize acquisition, plotting and user controls, the *PhysioKit* UI is implemented with a multi-threaded design. Figure 3A shows a *PhysioKit* interface, which was designed using Qt design tools (https://www.qt.io/product/ui-design-tools (accessed on 28 September 2023)), with controls to configure an experimental study alongside display of real-time signal visualization. Acquired raw data are stored locally and appropriate signal conditioning is applied for plotting each physiological signal, while the quality of the acquired signals is assessed in real-time using a novel 1D-CNN-based signal quality assessment module (Section 3.2.2). In addition, the UI enables options for different visual presentations of biofeedback (Section 3.2.4).

#### 3.2.2. Signal Quality Assessment Module for PPG

The signal quality assessment (SQA-Phys) module of *PhysioKit* extends 1D-CNN approach [57] that demonstrates the use of the 1D-CNN network for classifying the signal quality of PPG waveforms as “Good” and “Bad”. Contrary to classification, SQA-Phys introduces a novel task in which the ML model is trained to infer signal quality metrics for the entire length of the PPG signal segment. For this, we implement encoder–decoder architecture with 1D-CNN layers that generate high temporal resolution signal quality vector. Figure 4 compares the commonly used machine learning architecture with the proposed architecture, and highlights how the inferred outcome of SQA-Phys differs from the classification task as implemented by existing methods.

To train the model, we used in-house collected data acquired using the Infiniti Procomp and Empatica E4 wristband PPG sensors. This dataset, which includes 170 recordings of PPG signals (5 min) from 17 participants, was manually labeled for the signal quality. Signal quality labels for training were marked for 0.5 s of segment, without overlap. The temporal resolution of SQA-Phys inference was maintained as the same as the training labels (i.e., 0.5 s), as it is an optimal balance between dense temporal resolution (i.e., per sample inference) and classification. The optimal temporal resolution of 0.5 s can significantly reduce the computational complexity in comparison to the per sample semantic segmentation approach as proposed in a recent work [76].

We used a signal segment of 8 s as an input to the model and trained it with the batch size of 256 along with an Adam optimizer. The training and validation split was made based on participant IDs, ensuring that the signals of same individual were not represented in both the training and validation sets. Our validation on a novel signal quality assessment task shows 96% classification accuracy with an inference vector having 0.5 s temporal resolution, whereas the SOTA approach yielded 83% classification accuracy. The SQA-Phys is integrated with the *PhysioKit* to present signal quality assessment with optimal temporal resolution. The same is also stored alongside the data to provide signal quality annotation, thereby indicating the clean and noisy segments of the signals to researchers. Though SQA-Phys is currently implemented for PPG signals, it can be easily extended to other physiological signals.

#### 3.2.3. Configuring the UI for Experimental Studies

Using experiment and software configuration files (Figure 3B,C), the UI adapts to the sensor configuration as well as researcher’s data collection protocol. The experiment configuration file (Figure 3B) enables users to set up their study protocol by defining the experimental conditions, the acquisition duration for each condition, the required physiological sensors, and the directory path where the acquired data will be saved locally. In contrast to storing data on cloud servers, as most commercial devices do, storing data locally allows researchers to have complete control over the data. In addition, the experiment configuration file allows the number of channels to be selected for real-time plotting of acquired physiological signals. While the maximum number of channels is limited by the number of analog channels on the microcontroller board, a maximum of four channels can be selected for real-time plotting. The acquisition duration for each experimental condition is defined by “max_time_seconds” (see Figure 3B) when “timed_acquisition” is set to “true”. However, when the “timed_acquisition” field is set to “false”, the UI ignores “max_time_seconds”, allowing data acquisition to continue until the user manually stops it.

The software configuration file (Figure 3C) allows users to configure acquisition parameters, such as sampling rate and baudrate for serial data transfer from the micro-controller to the computer. In addition, users can acquire physiological signals simultaneously from multiple users, with each user connected through respective sensing and signal acquisition unit. Here, different computers running the UI would communicate using TCP/IP messaging in order to synchronize the acquisition from the different sensing and signal acquisition unit. This setup requires each computer to be accessible with an IP address, either on a local intranet or remotely using a virtual private network (VPN).

To enable multi-user synchronization, one computer is configured as a TCP server while the others are TCP clients. Server and client roles are specified using the software configuration file. While the server is configured to accept requests from any IP address, the configuration file at the client end is required to specify the server IP address. In these settings, the live acquisition is initiated on all nodes. The client UIs remain idle until the server triggers the synchronized recording by broadcasting a TCP message. The TCP/IP-based messaging can be further extended to the stimulus presentation software (not part of *PhysioKit*) for synchronized delivery of the study intervention. Figure 5 provides an overview of the multi-user setup.

Lastly, HCI studies oftentimes require capturing asynchronous events during experiments for qualitative and quantitative analyses. To address this need, we designed a simple way for users to mark asynchronous events directly in the *PhysioKit* UI by activating and deactivating the marking function. This function is associated with an event code to enable marking for different types of events during data acquisition. The acquired data, along with the signal quality assessment and event markings, are stored in a comma-separated value format (CSV) for easy access and further analysis.

#### 3.2.4. Support for Interventional Studies and Biofeedback

*PhysioKit* supports real-time computing of physiological metrics, which allows adapting interaction for interventional studies. One of the most widely researched interventional study types involves using biofeedback [4,77,78]. Using the experimental configuration file, researchers can specify a physiological metrics to be used for dynamic biofeedback visualization. Physiological metrics are computed in real-time with a provision for researchers to set the window length and step interval (see Figure 3B). *PhysioKit* currently provides options for basic biofeedback visualization using geometric shapes that vary in size and color. However, the mapping implementation can be easily adjusted to include different biofeedback modalities, such as auditory or haptic, according to the study requirements.

#### 3.2.5. Data Analysis Helper

The software layer also facilitates visualization and analysis of the data acquired using the *PhysioKit* by integrating NeuroKit2 [72] library. Jupyter notebooks are provided with the repository to illustrate loading, pre-processing and analysis of the acquired PPG, EDA and RSP signals. *PhysioKit* further supports batch processing of entire dataset acquired from multiple participants for a specific study. To offer flexibility for this analysis, a separate configuration file is provided to specify the key analysis parameters; these include the sampling rate, window length (seconds), overlap (seconds) and a list of physiological metrics to compute for PPG, EDA and RSP signals. Batch processing generates a spreadsheet as well as a NumPy [79] dictionary consisting of computed physiological metrics for all participants, and as per the specified analysis parameters. While both the spreadsheet and the NumPy dictionary comprise the identical analysis data, the former provides an easily accessible format for researchers with less familiarity with programming, and the latter immensely benefits researchers who would like to perform customized analysis in Python environment. The data are organized with participant ID and experimental conditions, as well as participant groups (if specified). The computed metrics are validated as per the normal healthy range for respective metrics. However, this can be adjusted by researchers using the configuration file. Lastly, if the signals were acquired from multiple sensors (e.g., PPG signal from finger and ear) to mitigate noise artifacts, the analysis can be configured to inspect and compare the signal quality for each window segment and ultimately select the one with the highest signal quality.

## 4. Evaluation

### 4.1. Study 1: Performance Evaluation

The first study focuses on validating the performance of *PhysioKit* in extracting heart rate and heart rate variability (HRV) data from two PPGs on different body parts (finger and ear), given our primary focus on the most widely available channel. Our data collection protocol was designed to ensure high variances in physiological patterns for fairer validation (e.g., a narrowed range of HRV values in a dataset tends to lead to high accuracy). We used the Procomp Infiniti System (https://thoughttechnology.com/procomp-infiniti-system-w-biograph-infiniti-software-t7500m/ (accessed on 28 September 2023)) as a reference system (at 256 Hz), as it has been widely used by researchers in both clinical and non-clinical studies. In order to assess level of movements that can affect the signal quality, we also video-recorded each session (at 30 fps).

#### 4.1.1. Data Collection Protocol

The study followed a methodological assessment protocol with the objective of inducing variations in physiological states, as well as moderating movement. Each participant experienced four conditions, including (a) a controlled, slow breathing task, (b) an easy math task, (c) a difficult math task, and (d) a guided head movement task, as depicted in Figure 6. While a higher agreement between the test device and the reference device could potentially be achieved in the absence of these variances, it could lead to misleading results.

Cognitively challenging math tasks with varying degrees of difficulty levels were chosen, as these have been reported to alter the physiological responses [80,81]. Furthermore, as wearable sensors are less reliable under significant motion conditions [82], we added an experimental condition that involved guided head movement. The PPG sensor on the ear remained relatively stable under all conditions, except during head movement, which provided us with the opportunity to investigate the impact of movement on signal quality at different sensor sites under varying motion conditions. Each condition lasted for 3 min, with 1 min of rest after each condition. To randomize the sequence of conditions, we inter-changed “A” with “D” and “B” with “C”. The study protocol was approved by the ethics committee of University College London Interaction Centre.

#### 4.1.2. Participants and Study Preparation

Three physiological signals (PPG, EDA, and RSP) were collected from 16 participants recruited through an online recruitment platform for research. All participants reported having no known health conditions, provided informed consent ahead of the study, and were compensated for their time following the study. After being welcomed and briefed, participants were asked to remove any bulky clothing (e.g., winter coats, jackets) and seated comfortably in front of a 65 by 37 inch screen, where they were fitted with both Infiniti and *Physiokit* sensors. Respiration belts from both systems were additionally worn just below the diaphragm, one above the other without overlapping. One *PhysioKit* PPG sensor was attached to participants’ left ear with a metal clip, and the second was strapped around the participant’s middle finger of their non-dominant hand with a velcro strap along with the Infiniti PPG. Extra EDA sensors from both *PhysioKit* and Infiniti systems were placed on the index and ring fingers of the same hand without overlapping. Of the 16 participants, one was excluded from the analysis due to the incorrect fit of a PPG sensor.

#### 4.1.3. Data Analysis

From 15 participants and four different experimental sessions, 60 pairs of PPG signals from each system (*PhysioKit* and Infiniti) were prepared to evaluate the system performance in extracting heart rate and heart rate variability data. Data analysis was performed using the data analysis module of *PhysioKit*, as described in Section 3.2.5. Pre-processing steps and signal quality analysis were uniformly executed for PPG signals acquired from *PhysioKit* and the reference device. A band-pass filter (0.7–4.0 Hz) of the third order was applied to PPG signals which were then processed to derive HR and HRV metrics. We applied windowing with a window size of 30 s [83] and a step interval of 10 s to calculate HR. As the HRV extracted from the PPG signal is referred to as pulse-rate variability (PRV) [10], in the following text, we mention it as PRV. For extracting PRV metrics, the window size was set to 120 s [84], with a step-interval of 10 s. In this work, our goal was to validate PRV metrics with the reference device, rather than validate PRV with HRV metrics, which are typically derived from the ECG signal. Among different PRV metrics, we selected pNN50, which provides a proportion of the successive heartbeat intervals exceeding 50 ms.

For fair evaluation, we used existing relative power signal quality index (pSQI) as described in [61]. pSQI was computed for each windowed segment from raw signals. Evaluation was conducted on PPG signal segments not affected by artifacts. For this, a threshold of 40% pSQI was applied to the PPG signals acquired using the reference device. For the segments of PPG signals of the reference device with less than 40% pSQI, corresponding segments from the *PhysioKit* were also discarded, with the assumption that the PPG signals from both devices are equally affected by motion artifacts.

#### 4.1.4. Results

Bland–Altman scatter-plots were used to assess the agreement between the *PhysioKit* and the reference device (see Figure 7) [85,86]. These plots provide a combined comparison for all experimental conditions for HR (Figure 7A,C) and HRV metrics (Figure 7B,D). Since PPG signals were acquired from two different sites—finger and ear—we present the comparison for these two sensor sites separately.

Table 2 reports a detailed evaluation across each experimental conditions for both finger and ear sites. Both HR and PRV metrics from *PhysioKit* show a high correlation, as measured with Pearson correlation coefficient (r), and lower difference, as measured with root mean squared error (RMSE), mean absolute error (MAE) and standard deviation of error (SD). There is a higher correlation for HR (bpm) between *PhysioKit* and the reference device (ear: *r* = 0.97, finger: *r* = 0.97) than for PRV (pNN50) (ear: *r* = 0.87, finger: *r* = 0.89), which aligns with results from earlier studies listed in Table A1 in Appendix A.

For PRV, however, the agreement with the reference device during the difficult math task (ear: *r* = 0.64, finger: *r* = 0.75) and the face movement task (ear: *r* = 0.88, finger: *r* = 0.92) is lower than in the baseline. We can attribute this decrease in performance of PRV metrics for cognitively challenging and movement conditions to motion artifacts. Earlier studies involving commercial wearable PPG devices have also reported similar decreases in PRV accuracy [87].

In Figure 8A, we compare the SQI for PPG signals from both *PhysioKit* and the reference device under different experimental conditions. *PhysioKit* demonstrates consistent signal quality across different experimental conditions, while the reference device shows higher variance across each experimental condition. For *PhysioKit*, it can also be observed that the signal quality of the PPG sensor placed on the ear is higher than that of the finger for all conditions except face movements. Figure 8B compares the magnitude of facial movements across different experimental conditions, which is computed as a standard deviation of change in inter-frame rotation angle. A Wilcoxon Signed-Rank test found that motion increased significantly during math tasks and face-movement conditions from the baseline condition. From Figure 8A,B, it can be inferred that, under normal cognitive tasks not involving voluntary facial movement, the ear is less affected by motion artifacts.

### 4.2. Study 2: Usability Analysis

#### 4.2.1. Use Cases

To demonstrate how *PhysioKit* can be applied in practice, *PhysioKit* was distributed to 10 different small-project teams (N = 44 members in total) from the department to use for their own research purposes for four to six weeks. In Table 3, we tabulate several examples of projects that made use of the toolkit for data acquisition as well as data analysis. Group members from each of these projects received a 1 h hands-on tutoring and spent four to six weeks utilizing the toolkit. Below, we summarize these projects by categorizing them into interventional and passive contexts (Section 2.1).

##### PhysioKit for Interventional Applications

Several groups explored using *PhysioKit* for developing affect recognition systems. For instance, some groups developed adaptive games that adjusted the level of gameplay difficulty by assessing acute stress using a combination of HR, HRV (e.g., pulse rate, pulse amplitude, interbeat interval), skin conductance and breathing rate. Another group examined how physiological responses from PPG and EDA could be mapped to a valence-arousal model to assess reactions to music. Other projects also examined the effects of biofeedback and social biofeedback visualizations of HR to overcome stressful scenarios (e.g., oral presentations) and promote mindfulness [4].

##### PhysioKit for Passive Applications

Project groups took advantage of *PhysioKit*’s diverse hardware and software functions to develop passive applications. For instance, a project that mapped stress during a virtual task assessed from an ear PPG showed that signals from this location were less subject to motion artifacts. Team members from a different group used *PhysioKit*’s event-marking function to generate a dataset for an affective music recommender system. Lastly, another research project team used the provision of synchronized acquisition for multi-user scenario to examine similarities in physiological responses during remote virtual reality experience.

#### 4.2.2. Data Collection

To gain insights on the usability of *PhysioKit*, we designed a questionnaire in Qualtrics that included three parts. The first section focused on obtaining demographic data and information on participants’ prior experience with physiological sensing and toolkits. For the second part of the survey, we designed a modified version of the Usefulness, Satisfaction and Ease of Use questionnaire (USE; [88,89]) to gather insights on usability. For each question in this part of the survey, participants had five choices as follows: 1 (Strongly Disagree), 2 (Somewhat Disagree), 3 (Neither Agree Nor Disagree), 4 (Somewhat Agree) and 5 (Strongly Agree). The final section asked optional open-ended questions regarding participant’s favorite aspects, and suggestions for improvements and novel features.

One person representing each of the 10 project teams participated in completing the questionnaire (five female, five male; 18 to 34 years old), which took an average of 22.5 min to complete. Participation was voluntary, and no identifying information was collected. All participants had either completed a postgraduate degree in computer science or were currently in the process of completing one. Seven participants had used consumer-grade wearables (e.g., Apple Watch, FitBit Sense 2, Garmin, Withings Steel HR, Google Watch) either daily (N = 5) or occasionally (N = 2), while three had never used one before. In terms of previous experience with open source micro-controller kits, six participants had used them for a previous project or class (i.e., Arduino, Raspberry Pi, micro:bit Texas Instruments device), while four participants had no previous experience with micro-controllers.

#### 4.2.3. Results

##### Usefulness and Ease of Use

Everyone who completed the survey acknowledged that *PhysioKit* was useful for their projects, with most strongly agreeing (N = 7). All participants also agreed that it gave them full control over their project activities (“Strongly agree”: N = 5) and facilitated easy completion of their project tasks (“Strongly agree”: N = 4). Several participants with limited computing experience (N = 3) considered using other physiological systems (e.g., Empatica, Apple Watch, fnv.reduce), but ultimately chose *PhysioKit* because it gave them full control over data and signals processing and setup.

Nearly all participants found that the provision of raw data were the most useful aspect of *PhysioKit* (N = 9), while almost all (N = 8) found that the ability to extract features, control data acquisition, and access supported data analysis was important. Also, having the flexibility to adapt *PhysioKit* to different experimental protocols was highly valued (N = 7). Participants found Physiokit easy to use (*M* = 4.30, *SD* = 0.675) and quick to set up (*M* = 4.00, *SD* = 0.943). They also appreciated that it enabled flexible configurations (*M* = 4.40, *SD* = 0.699).

##### Learning Process

Most participants learned to use *PhysioKit* quickly (*M* = 3.90, *SD* = 0.568) and with different sensor configurations for a diverse range of study designs (*M* = 3.90, *SD* = 0.568). Once they learned how to use the toolkit, everyone found it easy to remember how to use (*M* = 4.20, *SD* = 0.422), regardless of computing their prior experience.

##### Satisfaction

All participants were satisfied with the way *PhysioKit* worked (*M* = 4.20, *SD* = 0.422) and would recommend it to colleagues (*M* = 4.10, *SD* = 0.316). Many also found it essential for the completion of their projects (*M* = 4.60, *SD* = 0.699) and most would prefer to use it over other physiological systems for future projects (*M* = 3.50, *SD* = 0.527).

##### Open-Ended Questions

When participants were asked what they appreciated about *PhysioKit*, one person with limited programming experience responded: “*It’s easy to understand and user-friendly for people without a coding foundation*” (P3). People with high computing proficiency also found *PhysioKit* quick to setup, well-organized, simple and flexible to use. Lastly, participants left comments encouraging the further distribution of *PhysioKit*: *“Promote it, make it accessible to more people, [help them] understand the difference between using this product and using physiological sensors directly.”* (P3).

## 5. Discussion

In this section, we discuss our main findings and show how *PhysioKit* relates to and builds upon existing research.

### 5.1. Unique Propositions of PhysioKit

*PhysioKit* is a fully open source toolkit for physiological computing that implements an Arduino-based sensing and signal acquisition layer, offering researchers flexibility to configure one or more Arduino-compatible physiological sensors (PPG, EDA and RSP). The simple and user-friendly interface of the software application layer is effective in streamlining the physiological data acquisition for a variety of applications, including those involving biofeedback. Its provision to synchronize data acquisition for remotely located users is enabler for conducting remote studies that require acquiring physiological signals. Furthermore, it is cumbersome to manually inspect segments of acquired signals, specifically in case of long acquisition duration and high number of participants. SQA-Phys introduced in this work can be applied both in real-time and as a post-processing step to automate the quality assessment of PPG signals, significantly reducing efforts of researchers. Furthermore, access to raw data not only gives HCI researchers ownership and control over the data, but also enables computational research to develop robust algorithms for handling real-world environments. Taken together, these aspects all contribute to enhancing the usability of the toolkit for researchers.

The overall hardware cost of the *PhysioKit* is well below the least expensive of the commercial sensors mentioned in Table A1. For instance, with a setup of one Arduino Uno and one PPG sensor, the cost is less than USD 50. For the collection of sensors used in this work (i.e., two PPG sensors, one EDA sensor and a one RSP sensor), it amounts to less than USD 200, which is still less than the average cost of commercially available sensors in Table A1. The number of sensors that can be connected simultaneously is only limited by the number of analog input channels of the chosen Arduino board. The repository provides Arduino programs of up to four simultaneous sensors, though researchers can easily extend it to support higher numbers of sensors. The corresponding changes to configure the software application layer are limited to specifying the number and types of sensors in the experiment configuration file, making it a highly efficient and cost-effective solution to address different research needs.

### 5.2. Evaluating the Validation of PhysioKit

The validation study in Section 4.1 highlights very good agreement between *PhysioKit* and the gold standard for HR. The PRV metrics also show good agreement during the baseline condition, while there is acceptable agreement during experimental conditions involving significant movement. This difference between performance related to HR and PRV can be explained by the findings of a recent study that assessed the validity and reliability of PPG derived HRV metrics, and found that PPG sensors are less reliable for HRV measurements [90]. However, it is worth mentioning that the same study found PPG sensors to be accurate for measuring HR. It is also noteworthy that the overall performance of *PhysioKit* shows better agreement with the gold standard compared to the performance of existing PPG-based commercial devices mentioned in Table A1. The performance of *PhysioKit* can be largely attributed to its sensing and signal acquisition layer, as well as the processing pipeline that includes signal quality assessment, filtering, and extraction of physiological metrics.

Regarding flexible configuration, the validation study results show similar performance of the finger PPG sensor and ear PPG sensor, suggesting that alternative sensor sites for PPG can be explored to achieve different research objectives and support accessibility. The slightly low signal quality (pSQI) observed for finger PPG sensor can be attributed to the possible voluntarily movement of finger leading to more frequent motion artifacts, whereas inability to ambulate ear lobe makes it more promising site for acquiring PPG signals. This interpretation is further supported by the lower pSQI of ear-PPG signals for an experimental condition involving guided face movement (Figure 8A).

### 5.3. Assessing the Usability of PhysioKit

Projects with both interventional and passive application types utilized *PhysioKit*, demonstrating its versatility in supporting data acquisition in different settings. The usability survey results (see Section 4.2) highlight the usefulness, learning experience and favorable aspects of *PhysioKit*. Participants agreed that *PhysioKit* was essential for their projects because it provided them with control and flexibility over tasks and allowed them to accomplish what they wanted to do. While access to raw data were valued as the most important aspect of the toolkit, they also appreciated having control over data acquisition, feature extraction and data analysis. Participants felt *PhysioKit* was easy to use and seemed enthusiastic about using *PhysioKit* for future projects, as well as sharing it with colleagues and friends because of its open-source features.

### 5.4. Limitations and Future Work

In this work, we implemented PPG, EDA and RSP sensors. However, future work could explore integrating other contact-based physiological sensors commonly used in HCI research (e.g., EMG, ECG and EEG) [1,91]. The validation study [92,93,94] and signal quality assessment module (SQA-Phys) mentioned in this work currently only apply to data from PPG sensors. However, these can also be extended to other contact-based physiological sensors, such as the ones previously mentioned. While *PhysioKit* offers flexibility in choosing any physiological sensor compatible with Arduino, it does not introduce new sensor hardware. Therefore, in-the-wild scenarios such as physical activity were not evaluated in this work, since the challenges offered by such scenarios are associated with the sensor hardware and fitment.

The objectives of making *PhysioKit* an open-source toolkit are two-fold: (i) offering a flexible and cost-effective solution for research community, and (ii) leveraging the contributions from research community in addressing the existing limitations of *PhysioKit* towards introducing support for additional physiological sensors and biofeedback modalities, as well as extending the validation study for range of applications, sensor modalities and higher numbers of participants. The latter objective is expected to be achieved as a future work. Currently, the signal quality assessment module (SQA-Phys) mentioned in this work is limited to assess the quality of PPG signals, which could be extended to signals acquired using other sensors. Additionally, although the existing implementation of SQA-Phys provides high temporal resolution, its outcome is restricted binary classification at a given time-point. By using an appropriate activation function (e.g., ReLU) at the final layer of the 1D-CNN architecture, as well as replacing the classification loss with the regression loss function, it is possible to obtain a continuous score from 0 to 1 to gain better insights into the signal quality, which will be addressed as a future work.

While one project used event-marking function of *PhysioKit* to build a dataset to train affective music recommender system, no projects required performing data analysis based on the event-marking labels. In future work, this aspect of the toolkit can be further explored and validated. The hands-on training for groups participating in the usability study was provided owing to unavailability of elaborate installation and usage instructions for the toolkit at early development stage. The published repository of *PhysioKit* offers easy installation steps, along with detailed usage instructions. Furthermore, as a future work, tutorials for different application types will be made available at the repository homepage.

Our future implementation plan further considers implementing a processing pipeline for contact-less sensing methods, such as an RGB camera-based remote PPG [95,96] and a thermal infrared sensing pipeline, including optimal quantization [9] and semantic segmentation [97] for extraction of breathing [9] and blood volume pulse signals [56,98,99]. We also aim to further enhance the accessibility of the sensor interface (hardware) of *PhysioKit* for its use in real-world scenarios.

## 6. Conclusions

This paper introduced *PhysioKit*, an open-source and cost effective physiological computing toolkit that streamlines physiological signal acquisition and analysis for various HCI studies and applications. Uniquely, *PhysioKit* can synchronize signal collection in multi-user studies with both co-located as well as remote users. The evaluation study on heart rate and pulse rate variability measurements demonstrated the performance of *PhysioKit* compared to the reference system. The comparable signal reliability from the finger and ear PPG channels further indicates the possibility of enhancing accessibility to support participants with certain physical impairments (e.g., wheelchair users). Further, the usability study highlighted the usefulness, satisfaction and ease of use associated with PHysioKit, emphasizing its positive impact in various application scenarios. *PhysioKit* provides further useful features for researchers and practitioners: visual biofeedback, which can be extended to other forms of biofeedback such as audio and haptics; and machine-learning-driven signal quality assessment that can significantly reduce efforts and time on manual signal inspection (for discarding noisy signal segments). Also, raw physiological data are stored in an accessible format, which can be organized as per the study protocol and participant ID and can foster opportunities for researchers to easily apply state-of-the-art analysis. *PhysioKit* thus provides a one-stop approach that supports physiological sensing, data acquisition and computing supporting for a broad spectrum of studies and HCI applications. 

## Figures and Tables

**Figure 1 sensors-23-08244-f001:**
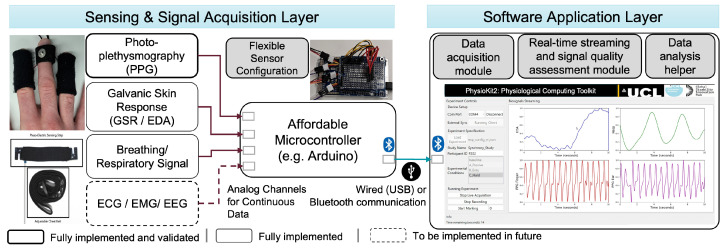
*PhysioKit* is a novel physiological computing toolkit which is open-source and cost-effective. Its sensing and signal acquisition layer provides flexibility in using low-cost physiological sensors, and the software application layer enables data acquisition, visualization, and real-time ML-based signal quality assessment, while supporting passive as well as interventional studies for single and multi-user settings.

**Figure 2 sensors-23-08244-f002:**
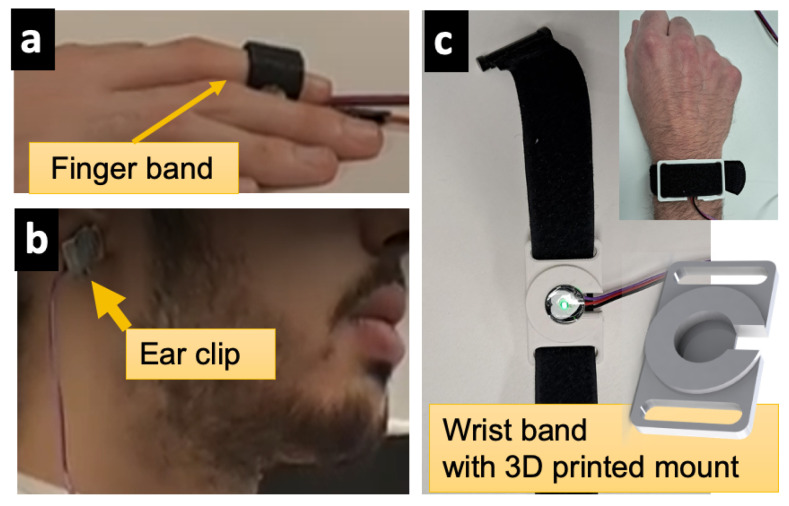
Flexible PPG setups available in *PhysioKit* to enhance accessibility.

**Figure 3 sensors-23-08244-f003:**
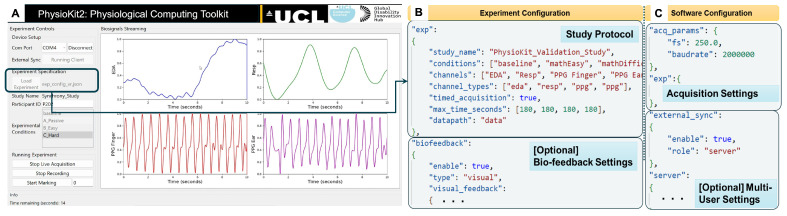
(**A**) User-interface for Physiological Computing Toolkit; (**B**) configuration file for specifying experiment protocol; and (**C**) configuration file for controlling data acquisition.

**Figure 4 sensors-23-08244-f004:**
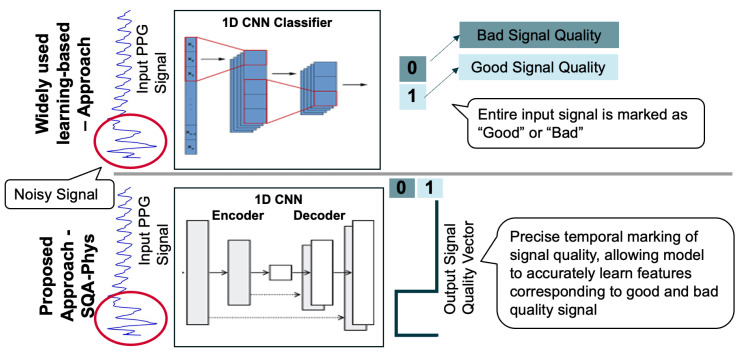
SQA-Phys: 1D-CNN-based encoder–decoder architecture for high temporal precision signal quality assessment.

**Figure 5 sensors-23-08244-f005:**
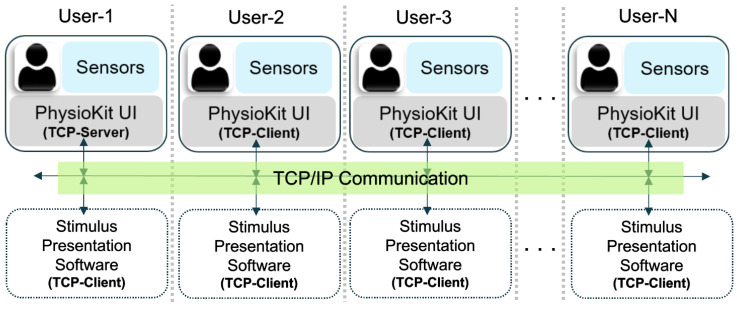
Multi-user setup of *PhysioKit* for synchronized data acquisition and stimulus presentation.

**Figure 6 sensors-23-08244-f006:**
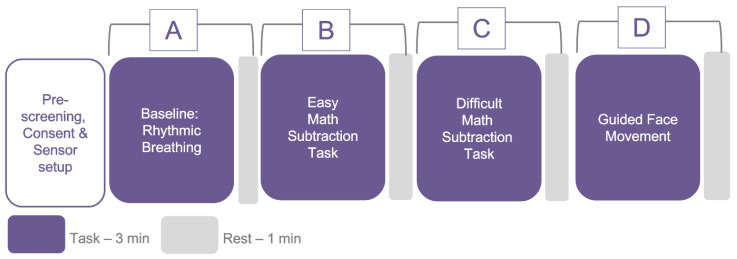
Data collection protocol for technical evaluation of *PhysioKit*. Four conditions that comprise the protocol are (A) baseline, (B) easy math task, (C) difficult math task, and (D) guided head movement task.

**Figure 7 sensors-23-08244-f007:**
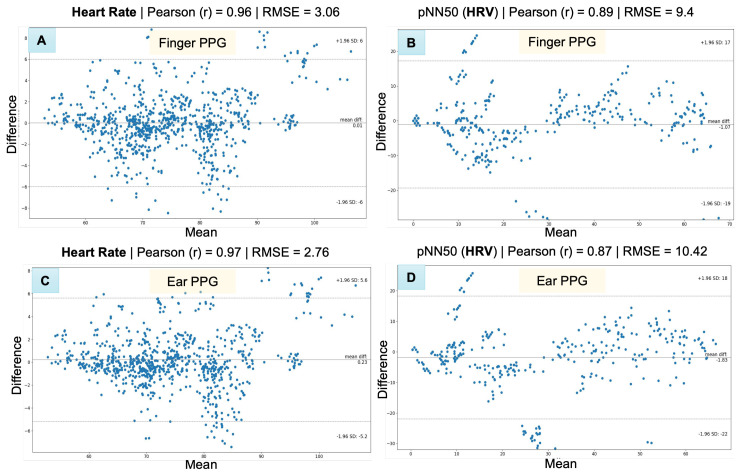
Bland−Altman scatter plots to compare heart−rate (beats per minute) (**A**,**C**), and pulse−rate variability (pNN50) (**B**,**D**) with the reference device.

**Figure 8 sensors-23-08244-f008:**
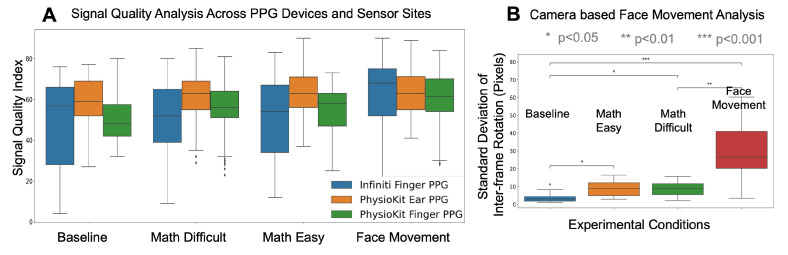
(**A**): Comparison of pSQI for PPG signals of *PhysioKit* and the reference device, under different experimental conditions; (**B**): experimental condition-wise comparison of facial movement.

**Table 1 sensors-23-08244-t001:** *PhysioKit*: specifications for sensing and signal acquisition layer along with package description.

Sensing and Signal Acquisition Layer Specifications	
**Parameter**	**Specifications**
Sampling rate	10–10,000 samples per second
ADC	10/12 bit
Vref	3.3V/ 5V
Baudrate	9600–2,000,000
Data transmission mode	USB/ Bluetooth
No. of channels	1—Max supported by specific Arduino board
**Installation and package contents**	
Installation of the interface	pip install PhysioKit2
GitHub repository	Repo link https://github.com/PhysiologicAILab/PhysioKit (accessed on 28 September 2023)
Configuration files	Download path https://github.com/PhysiologicAILab/PhysioKit/tree/main/configs
	(accessed on 28 September 2023)
Codes to program Arduino	Download path https://github.com/PhysiologicAILab/PhysioKit/tree/main/arduino
	(accessed on 28 September 2023)
3D-printable model for a wristband case	Download path https://github.com/PhysiologicAILab/PhysioKit/tree/main/CAD_Models
	(accessed on 28 September 2023)

**Table 2 sensors-23-08244-t002:** Evaluation of *PhysioKit*: comparison of heart-rate (beats per minute) and pulse-rate variability (pNN50) with the reference device. Evaluation metrics include root mean-squared error (RMSE), mean absolute error (MAE), standard deviation of error (SD) and Pearson correlation coefficient (r), which are mentioned separately for each experimental condition as well as for all conditions combined.

Metrics	PPG Site	Experimental Condition	RMSE	MAE	SD	Pearson (r)
**Heart** **Rate** **(beats** **per minute)**	**Finger**	Baseline	3.63	2.43	3.38	0.96
		Math—Easy	2.08	1.44	2.02	0.98
		Math—Difficult	2.41	1.75	2.41	0.98
		Face Movement	2.17	1.49	2.09	0.98
		All sessions combined	2.65	1.78	2.65	0.97
	**Ear**	Baseline	3.71	2.46	3.41	0.95
		Math—Easy	1.0	0.73	0.92	1.0
		Math—Difficult	2.08	1.46	2.07	0.98
		Face Movement	2.11	1.40	2.05	0.98
		All sessions combined	2.45	1.53	2.43	0.97
**Pulse** **Rate Variability** **(pNN50)**	**Finger**	Baseline	4.31	3.46	4.28	0.98
		Math—Easy	11.33	6.54	10.6	0.91
		Math—Difficult	12.11	9.53	12.09	0.75
		Face Movement	8.26	5.66	8.25	0.92
		All sessions combined	9.4	6.26	9.34	0.89
	**Ear**	Baseline	4.36	3.78	4.21	0.99
		Math—Easy	9.85	5.75	9.27	0.91
		Math—Difficult	15.26	11.13	15.12	0.64
		Face Movement	9.38	7.27	9.32	0.88
		All sessions combined	10.42	7.02	10.26	0.87

**Table 3 sensors-23-08244-t003:** Project teams specific application cases of *PhysioKit*.

Application	Application Type ${}^{†}$	No. of Members	Project Duration
Using physiological reactions as emotional responses to music	Interventional	6	4 weeks
Emotion recognition during watching of videos	Interventional	6	4 weeks
Using artistic biofeedback to encourage mindfulness	Interventional	1	6 weeks
Using acute stress response to determine game difficulty	Interventional	7	4 weeks
Generating a dataset for an affective music recommendation system	Passive	7	4 weeks
Adapting an endless runner game to player stress levels	Interventional	8 7	4 weeks
Influencing presentation experience with social biofeedback	Interventional	1	6 weeks
Mapping stress in virtual reality	Passive	1	6 weeks
Assessing synchronous heartbeats during a virtual reality game	Passive	2	6 weeks

^†^ See Section 2.1.

## Data Availability

Data are unavailable due to privacy or ethical restrictions.

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
