# Peer review of "PhysioKit: An Open-Source, Low-Cost Physiological Computing Toolkit for Single- and Multi-User Studies"

_sensors, 2023, doi:10.3390/s23198244_

Round 1

Reviewer 1 Report

The novelty and contribution of this paper cannot be found so easy inside the text, and therefore some changes are needed. The research approach is very weak, and the findings need to be strengthened. Here are my comments on improving the manuscript:

1. Overall:

a)       Why do the authors conduct this study? The research contributions are weak. Please kindly explain.

b)      The research structure is not appropriate for a scientific article, e.g., research questions are missing from the introduction, results and conclusion as well as the discussion is not written in an “appropriate” manner. Please update.

c)       Please consider how to effectively integrate some review papers and update.

2. Introduction:

a)       Research questions, that drive the paper, should be built in the introduction from an ongoing and pertinent bibliography (up to 2022-23) and these should be of global interest and not focused on a particular local problem. Identifying a research gap is the most important by indicating in-text some newer references that are significant to your particular field of research.

b)      Due to a weak contribution, please pay attention to addressing new research gaps and then emphasize why the authors do this study.

3. Discussion:

a)       The proposed reason for conducting such a study is weak due to a lack of explanation.

b)      The result analysis is poor and subjective due to a lack of contributions and thorough discussion. Please rewrite it and consider providing a new discussion section to provide significant criticism and research limitations.

c)       Authors should answer your research question in the conclusions and discussion. Please provide a reasonable need to read your work’s results than previous ones or simply answer what we learned compared with current, significant research (up to 2022 should be your work’s “significance”).

d)      How general are your results and how do you believe that such findings have to be of global interest? Please relate these with your limitations and Discussion that is not exist. Why?

e)      Are there any points of view related to the consequences of this study’s limitations that may have an impact on their findings?

4.    Conclusions and limits are too short for such a study.

a)       Are there any points of view related to the consequences of this study’s limitations that may have an impact on their findings?

b)      Implications for practice and method are not provided.

-

Reviewer 2 Report

  • The article provides a large number of references, 

    however, the literature in recent years, especially in the 2022-2024 period, is too small to account for the recent relevant research results. The more recent literatures must be cited to improve the quality of the paper.
  •  
1. In Introduction, author described past work, but little comment on the contribution and shortcoming. Author need to provide critical comments. Please highlight how the work advances or increments the field from the present state of knowledge and provide a clear justification for your work. 2. In the conclusion, please show how the work advances the field from the present state of knowledge. Please provide a clear justification for your work in this section. The conclusion section has to be rewritten doing an effort to remark the main findings rather than summarizing the article content.

Reviewer 3 Report

This manuscript proposes a toolkit for collecting physiological signals in user studies, which contains a hardware and a software layer.

The PPG sensor was evaluated in one main study with 15 participants, but also in 10 side projects.

Global comments :

Overall, I really like your proposition of a toolkit that can be easily adaptable for research needs to collect physiological signals in user studies. I conducted this kind of experiments myself and I experienced some of the challenges and issues you mention in your manuscript. I think it could be a great contribution to the scientific community, especially if it includes a machine learning module to predict the users' state in various use case scenarios or context. Many details are given and it reads well. The structure can be slighlty adapted (section 6 in Evaluation).

However, there are certain things to address before the manuscript can be published. The related work should be revised according to my suggestions, more details can be given when you present your toolkit, some statements need better justification, the discussion can be elaborated a bit, and a conclusion is missing. You'll find my detailed comments below. If these are addressed, I'm in favour of publishing the manuscript.

1 Introduction 

- The flow of the introduction lacks logic in my opinion. 

- L37-38 "PPG, EDA, and RSP are among the most prominent physiological sensing channels used in HCI research." : Reference for this statement ?

- L 37 to 45 are not necessary for the understanding in the introduction, and is more about related work. Ideas are a bit disconnected : you talk about education, then gamers, then emotional state, biofeedback, stress, etc... I know physiological sigmla sar eused in many field but it could be presented in a better way

- L46-62 : Very good ! I agree with the challenges the scientific community of this field is facing

2. Related Work

- L95-97 : the distinction of the two categories is interesting. Is there any previous work that proposed such disctinction ? Otherwise, please explain.

- Actually it is done later. Maybe you can move L114-121 before L98 ?

- Font in table 1 a bit small, please adress this. Also what do you call "Discontinued device" ? Please explain.

- L152 : typo, ) missing

- What about devices like Movisens ?

- For me, a section on sofwtare, platforms or libraries (e.g. Neurokit2 in Python) that already exist to collect, process, and analyze physiological signals is needed. Since the software layer is one of your contribution, I would expect a related work section on that (both commercial and academic solutions). Regarding hardware, I would also include a subsection on sensor propositions to collect users' physiological signals in various academic research fields.

3. Problem Statement and Challenges

- 3.1 : Assessing signal quality is definitely necessary

- 3.2 : Okay but is has already been mentioned in the introduction

- Overall, I think this section could be reduced and included in section 4 to introduce section 4 and explain what problem(s) your solution aims at solving

4. PhysioKit

- Are you planning to use Docker to make sure people don't have version problems and can deploy and use your solution on any device without problem ?

- 4.1 Is it working with any sensor that can be connected to an Arduino ? Aare they some tutorials how to plug sensors on Arduino for end users who don't know how to do it ? Or they should refer to the constructor instructions?

- I think providing the CAD model to print the PPG sensor housing is also very relevant.

- 4.2.2 : your proposition is intersting, not to classify the entire signal but rather chunks of the signals. Would it be possible to go further and do some regression (predict signal quality as a value between 0 and 1) rather than classification ?

- L299-300 : good !

-4.2.5 : I would expect more details here. What can we analyse ? Can we do compute and calculate features over different time windows ? With or without overlapping ? is there any feature implemented to calculate baseline-corrected features ? If not, mention in the limitation (7.4) as further features to develop in your toolkit.

5. Evaluation

Mention here that it is a preliminary evaluation as you have less than 20 participants. Same in section 6.

5.1.3. : 

- The analysis was done with your own PhysioKit platform ?

- Any justification for the choice of these time windows ?

- L419-424 : "No segments acquired using PhysioKit were discarded based on pSQI.". But how many from the reference signals ?

5.1.4. 

- Please cite make a reference to Figure 7. And explain how it should be interpreted in one or two sentence(s).

- I think the reference to figures are wrong in this section, please fix this.

- Looking at figure 8, it's seems strange to me that no segments calculated for data collected with PhysioKit sensors were discarded with the 40% threshold on pSQI0

6. Study 2

Section 6 should be 5.2 I guess ?

Participants were people from your academic institution I gues (colleagures from your lab or other professors). Then you should mention it in this section and/or in the limitations

6.1.1 : L490-492 : State the number of options participants had on Likert-scale (I guess it's 5).

6.2 Very interesting projects and use case scenrios, but I would present this earlier in section 6.

7. Discussion 

Overall, the discussion is bit short in my opinion. Section 7.2 could be further elaborated, and 7.3 is not very relevant according to me.

7.1. Though results presented are encouraging, I would talk about preliminary results here as the evaluation process was done with less than 20 participants.

L557-559 : Okay, but pQSI seems lower on the finger. Please mention and discuss this.

7.4 : There are other limitations to mention I guess : the small number of participants, test sensors from different constructors, replicate results with more participants, in other conditions, in more challenging contexts (e.g. physical activity, driving, etc..).

A conclusion is missing.

The level of English is quite good for a scientific article, although the introduction and related work could be a little more elaborate, with a better flow of ideas.

Round 2

Reviewer 1 Report

The authors have satisfactorily addressed all inquiries and integrated the necessary revisions into the manuscript. As a result, I am pleased to endorse the approval of this study for publication.

Reviewer 2 Report

Accept